# The *Yellow Stripe-Like* (*YSL*) Gene Functions in Internal Copper Transport in Peanut

**DOI:** 10.3390/genes9120635

**Published:** 2018-12-14

**Authors:** Jing Dai, Nanqi Wang, Hongchun Xiong, Wei Qiu, Hiromi Nakanishi, Takanori Kobayashi, Naoko K. Nishizawa, Yuanmei Zuo

**Affiliations:** 1Key Lab of Plant-Soil Interaction, MOE, Center for Resources, Environment and Food Security, College Resources and Environmental Sciences, China Agricultural University, Beijing 100193, China; daijing@cau.edu.cn (J.D.); wnqqnw13579@163.com (N.W.); qiuwe1279@sina.com (W.Q.); 2Graduate School of Agricultural and Life Sciences, The University of Tokyo, 1-1 Yayoi, Bunkyo-ku, Tokyo 113-8657, Japan; ahnaka@mail.ecc.u-tokyo.ac.jp (H.N.); annaoko@mail.ecc.u-tokyo.ac.jp (N.K.N.); 3Institute of Crop Science, Chinese Academy of Agricultural Sciences, Beijing 100081, China; xionghongchun@caas.cn; 4Research Institute for Bioresources and Biotechnology, Ishikawa Prefectural University, 1-308 Suematsu, Nonoichi, Ishikawa 921-8836, Japan; abkoba@ishikawa-pu.ac.jp

**Keywords:** copper excess, *AhYSL3.1*, transporter, peanut, transgenic plant, iron

## Abstract

Copper (Cu) is involved in fundamental biological processes for plant growth and development. However, Cu excess is harmful to plants. Thus, Cu in plant tissues must be tightly regulated. In this study, we found that the peanut Yellow Stripe-Like family gene *AhYSL3.1* is involved in Cu transport. Among five *AhYSL* genes, *AhYSL3.1* and *AhYSL3.2* were upregulated by Cu deficiency in peanut roots and expressed mainly in young leaves. A yeast complementation assay suggested that the plasma membrane-localized *AhYSL3.1* was a Cu-nicotianamine complex transporter. High expression of *AhYSL3.1* in tobacco and rice plants with excess Cu resulted in a low concentration of Cu in young leaves. These transgenic plants were resistant to excess Cu. The above results suggest that *AhYSL3.1* is responsible for the internal transport of Cu in peanut.

## 1. Introduction

Copper (Cu) is an essential micronutrient that plays key roles in multiple plant processes, including photosynthesis, respiration, carbon and nitrogen metabolism, oxidative stress, cell wall synthesis, and hormone perception [1,2]. Cu deficiency in plants often leads to stunted growth, chlorosis in young leaves, and defects in development [3]. In contrast, excess Cu can also be toxic to plants due to the production of highly toxic hydroxyl radicals by redox cycling between Cu(I) and Cu(II). Cu toxicity usually results in reduction of plant biomass, inhibition of root growth, chlorosis, bronzing, and necrosis [4]. Moreover, excess Cu in the environment exerts harmful effects on all living organisms [5]. Therefore, Cu homeostasis must be tightly regulated in plants, and indeed all organisms [6].

For maintenance of appropriate Cu levels in plant tissues, a sophisticated homeostatic network is controlled by various types of transporter proteins. The conserved Cu transporter (CTR) family proteins or CTR-like transporters (COPT) have been well characterized. In *Arabidopsis* (*Arabidopsis thaliana*), six CTR family members have been identified [7]. Among them, *COPT1* and *COPT2* restored the growth of a mutant yeast strain, while *COPT3* and *COPT5* partially rescued the defective growth of mutant yeast, as determined by a functional complementation assay [8]. The high-affinity copper transporter COPT1 plays an important role in Cu uptake and pollen development [9,10]. However, *COPT2* participates in the cross-talk between iron-deficiency responses and low-phosphate signaling [11]. The tonoplast-localized *COPT5* functions in inter-organ reallocation of copper ions from the root to reproductive organs [3] and is required for photosynthetic electron transport under Cu scarcity [12]. The plasma membrane transporter AtCOPT6 is responsible for Cu distribution and homeostasis during Cu limitation and excess [13,14]. The rice (*Oryza sativa*) CTR family consists of seven members. It was suggested that rice *COPTs* function alone or cooperatively to mediate Cu transport in various tissues [13,15]. *OsCOPT* expression is affected at the transcriptional level by multiple factors, including the levels of Cu and other bivalent cations [13].

The P-type heavy-metal ATPase (HMA) family proteins have also been implicated in the transport of Cu in plants. *Arabidopsis HMA1*, *HMA6*/*PAA1*, and *HMA8*/*PAA2* are required for Cu delivery in chloroplasts [16,17,18,19]. Moreover, *AtHMA5* is involved in Cu detoxification and tolerance in *Arabidopsis* [20,21]. In rice, the transcript level of the Cu^+^ transporter OsHMA5 was upregulated by excess Cu, but not by deprivation of Cu or other metals. Mutation of *OsHMA5* resulted in a decreased Cu concentration in the xylem sap and shoots, but increased Cu concentration in the roots, suggesting its role in Cu loading of xylem in roots [22]. In contrast, the OsHMA4 was proved to function to sequester Cu into root vacuoles [23]. The expression of *OsHMA9* was induced by a high concentration of Cu, Zn, and Cd, and it plays a role in the efflux of these metals from cells [24]. It was reported that the cucumber HMA5-like ATPases may contribute to vacuolar sequestration of Cu in vacuoles under Cu excess [25].

Nicotianamine (NA) is a key compound for metal transport [26,27]. It was reported that NA is required for delivery of Cu in the xylem [28,29]. The Yellow Stripe-Like (YSL) family proteins have been found to participate in Cu–NA transport. *Arabidopsis* YSL2 (AtYSL2) is suggested to transport both Fe(II)–NA and Cu–NA [30], although another report argues against transport of these substrates [31]. The *AtYSL2* transcriptional level is regulated by Fe and Cu [30]. Importantly, the *Arabidopsis* small ubiquitin-like modifier (SUMO) E3 ligase SIZ1, which regulates the expression of *AtYSL1* and *AtYSL3*, functions in excess Cu tolerance in plants [32]. Additionally, rice YSL16 is involved in the delivery of Cu–NA to developing young tissues and seeds through phloem transport. The expression of *OsYSL16* was upregulated by Zn and Fe deprivation, but not by Mn and Cu deficiency, in the roots. It was demonstrated by yeast expression assay that *OsYSL16* transports Cu–NA, but not ionic Cu, and the Cu-deoxymugineic acid (DMA) complex. Knockout of *OsYSL16* resulted in a significant reduction in Cu–NA translocation from older to younger leaves and from the flag leaf to the panicle [33]. The *osysl16* mutant showed low pollen germination which could be rescued by addition of Cu [34]. In addition to Cu translocation, OsYSL16 is also involved in Fe allocation in rice plants [35,36] by transporting Fe(III)-DMA, as indicated by a yeast complementation assay [36]. DMA is an Fe(III)-chelating molecule biosynthesized in graminaceous plants, which functions in Fe uptake from the soil and Fe translocation within the plant [37]. In our previous study, we identified five peanut *YSL* genes. Among these, the expression of only *AhYSL1* was induced by Fe deficiency in the roots. Yeast functional complementation indicated that *AhYSL3.1* transports not only Fe(III)–DMA, but also Fe(II)–NA, while *AhYSL1* specifically transports Fe(III)–DMA. It was implied that *AhYSL1* functions in Fe acquisition in the peanut maize intercropping system [38]. However, the functions of other peanut *YSL* genes remain unknown. In the present study, the peanut *AhYSL3.1* gene was characterized. *AhYSL3.1* is suggested to be responsible for internal transport of Cu in peanut.

## 2. Materials and Methods

### 2.1. Plant Materials and Growth Conditions

Peanut (*Arachis hypogaea* L. cv. Luhua 14) plants were grown in 5 L boxes with the following nutrient solution: 0.7 mM K_2_SO_4_, 0.1 mM KCl, 0.1 mM KH_2_PO_4_, 2.0 mM Ca(NO_3_)_2_, 0.5 mM MgSO_4_, 10 μM H_3_BO_3_, 0.5 μM MnSO_4_, 0.5 μM ZnSO_4_, 0.2 μM CuSO_4_, 0.01 μM (NH_4_)_6_Mo_7_O_24_, and 100 μM Fe(III)-EDTA. For metal treatments, the levels of the corresponding metals in the above nutrient solution were modified. Peanut plants were cultured in a greenhouse with aeration and 30 °C light/25 °C dark cycles under natural light conditions.

### 2.2. Quantitative Real-Time PCR

For analysis of *AhYSL* gene expression in response to metal treatments, roots or leaves of peanut plants subjected to Cu deficiency (–Cu), Mn deficiency (–Mn), Zn deficiency (–Zn), or excess Cu (25 μM Cu) in hydroponics for 5 days were harvested. For analysis of tissue expression of *AhYSL* genes, parts of roots, stems, young leaves, and old leaves from 7-day-old peanut plants were sampled.

The samples were subjected to total RNA extraction and then treated with RNase-free DNase I (Takara, Tokyo, Japan) to remove genomic DNA contamination [39]. First-strand complementary DNA (cDNA) was synthesized using ReverTra Ace reverse transcriptase (Toyobo, Tokyo, Japan) by priming with the d(T)_17_-adaptor primer. Quantitative real-time PCR was performed using the StepOnePlus™ real-time PCR system (Applied Biosystems, Foster City, CA, USA) and SYBR Premix Ex Taq (Perfect Real Time) reagent (Takara, Tokyo, Japan) using primers reported previously [38]. The PCR products were confirmed by DNA sequencing (3130 Genetic Analyzer, Applied Biosystems, Tokyo, Japan). The peanut *ubiquitin* gene [40] was used as an internal control.

### 2.3. Subcellular Localization of *AhYSL3.1*

The *AhYSL3.1* open reading frame (ORF) without a stop codon and containing *Xba*I and *Bam*HI restriction enzyme sites was amplified by PCR using the primers 5′-TCTAGAATGAATCCAATGGAAATCAAT-3′ and 5′-GGATCCCTTGGATGTAAAGAAGCT-3′. The PCR product was then subcloned into the CaMV35S-sGFP(S65T)-NOS3’ vector (kindly provided by Dr. Yasuo Niwa, University of Shizuoka, Japan) to generate the AhYSL3.1-sGFP construct. The AhYSL3.1-sGFP construct or the control CaMV35S-sGFP, together with 35S-DsRed (Clontech, Mountain View, CA, USA), was then bombarded into onion epidermal cells by DNA particle bombardment as described by Mizuno et al. [41]. After incubation in the dark for 16 h, fluorescent cells were imaged by confocal microscopy (LSM5Pascal; Carl Zeiss, Göttingen, Germany).

### 2.4. In Situ Hybridization

In situ hybridization was performed as described previously [39]. Peanut roots grown hydroponically were fixed in formalin-acetic acid-alcohol and embedded in paraffin. The root tissues were sectioned into 10 μm slices and mounted on slides. The 3′UTR fragment of *AhYSL3.1* was amplified using the primers: AhYSL3.1probe-F, 5′-TTTGCGATAGCAGCCAACTTGGTGAG-3′ and AhYSL3.1probe-R, 5′-AATTGTAGTTGCAAACTAGATACACTGATC-3′. After subcloning into the pCR^®^-Blunt II-TOPO^®^ vector (Invitrogen, Carlsbad, CA, USA), the plasmid was linearized using *Bam*HI. T7 RNA polymerase was used to generate sense and antisense probes, and the generated probes were labeled with digoxigenin-11-UTP (Roche, Mannheim, Germany). The slides were pretreated with proteinase K and subjected to acetylation and then hybridized for 16 h with the sense or antisense probe at 50 °C. After washing, the sections were incubated with anti-digoxigenin alkaline phosphatase conjugate (Roche) and stained with nitroblue tetrazolium/5-bromo-4-chloro-3-indolyl phosphate (Roche).

### 2.5. Functional Complementation in Yeast

The *AhYSL3.1* ORF with *Not*I and *Bam*HI restriction sites was amplified by PCR using the primers: YAhYSL3.1-F, 5′-TGCGGCCGCATGAATCCAATGGAAATC-3′; YAhYSL3.1-R, 5′-CCGCGGATCCCTACTTGGATGTAAAGAAGCT-3′. The PCR product was then subcloned into the pCR^®^-Blunt II-TOPO^®^ vector (Invitrogen) and confirmed by sequencing. After digestion, the construct was introduced into the yeast expression vector pDR195 [31] (kindly provided by Dr. Nicolaus von Wirén, University of Hohenheim, Germany) to generate the AhYSL3.1-pDR195 construct. The Cu-deficient yeast (*Saccharomyces cerevisiae*) mutant M10 was used for complementation assay. This *CTR1* mutant was kindly provided by Dr. Andrew Dancis (National Institutes of Health, Maryland, USA). The construct AhYSL3.1-pDR195 or empty vector pDR195 was transformed into the yeast strain M10 by the LiAc/SS-DNA/PEG method [42]. Serial dilutions of the transformed yeast cells of OD_600_ 1 to 0.001 were plated onto Synthetic Defined (SD) medium containing 9 μM Cu–NA or 10 μM CuSO_4_.

### 2.6. Generation of Transgenic Lines

The full-length *AhYSL3.1* ORF with *Xba*I and *Sac*I restriction sites was amplified using the following primers: 5′-TCTAGAATGAATCCAATGGAAATCAA-3′; 5′-GAGCTCCTACTTGGATGTAAAGAAGC-3’. The *GUS* gene in the E-90Ω plasmid [43] was replaced with the *AhYSL3.1* ORF. As shown in Appendix A, the resultant plasmid drives expression of *AhYSL3.1* under the control of the Fe deficiency-responsive *cis*-acting elements IDE1 and IDE2 connected to the −90/+8 region of the cauliflower mosaic virus 35S promoter, which drives preferential expression in vascular tissues [43,44], and the tobacco mosaic virus 5′ leader sequence for expression enhancement in the backbone of the pIG121Hm binary vector [45]. After introduction into *Agrobacterium tumefaciens* strain C58 by electroporation, the *Agrobacterium* carrying the above construction was transformed into tobacco (*Nicotiana tabacum* L. cv. Petit-Havana SR1) and rice (*Oryza sativa* L. cv. Tsukinohikari). The tobacco transformation was performed according to the procedure described by Helmer et al. [46] and the rice transformation by that of Hiei et al. [45].

### 2.7. Analyses of Transgenic Tobacco and Rice Plants

The transgenic lines were germinated on Murashige and Skoog (MS) medium containing hygromycin B (50 mgL^−1^). Non-transgenic seeds were germinated on MS medium lacking hygromycin B as a control. After 2–3 weeks of growth, the plants were transferred to the previously described nutrient solution [39] under a 14 h light/10 h dark cycle with 28/24 °C day/night temperatures and a light intensity of 300 μmol m^−2^ s^−1^ provided by reflector sunlight dysprosium lamps and 60% relative humidity. For tobacco, eighteen transgenic lines were generated. Two lines with higher expression of *AhYSL3.1*, T10 and T11, were selected for further analysis. Four replicates of the transgenic and non-transgenic tobacco plants were grown hydroponically for ~1 week and then treated without Cu and with 0.2, 25, or 50 μM Cu for 6 days. Roots, stems, young leaves, and old leaves were divided and harvested for metal content determination. The rice plants were cultured in normal nutrient solution for ~1 week and then treated with 100 μM Cu for 6 days. The young leaves, old leaves, and roots were then sampled. The above samples were dried for 2–3 days at 70 °C, and ~100 mg portions were then digested with HNO_3_-H_2_O_2_ in a microwave accelerated reaction system (CEM, Matthews, NC, USA). Metal concentrations were measured by inductively coupled plasma atomic emission spectroscopy (ICP-AES, OPTIMA 3300 DV, Perkin-Elmer, Norwalk, CT, USA).

### 2.8. Transmembrane Protein Structure Prediction and Statistical Analysis

Combined membrane protein topology and signal peptide prediction was conducted by the TOPCONS web server [47]. All data were analyzed using SPSS 23.0 software (SPSS Inc., Chicago, IL, USA). All of the means were tested using one-way analysis of variance at the 5% and 1% probability level.

## 3. Results

### 3.1. Expression of *AhYSL* Genes in Response to Metal Treatments

In a previous study, five *AhYSL* genes were identified in peanut, and their response to Fe deficiency was analyzed [38]. To determine whether these *YSL* genes function in transport of other metals, their expression in response to other metal treatments was analyzed. Among them, *AhYSL3.1* and *AhYSL3.2* were induced more than twofold in Cu-deficient roots, but not in Mn- and Zn-deficient roots, as shown in Figure 1A,C. *AhYSL3.1* mRNA levels were slightly increased by excess Cu, but were decreased by Mn and Zn deficiency in leaves, as shown in Figure 1B. In contrast, *AhYSL3.2* expression was suppressed by excess Cu in roots, as shown in Figure 1C, and Mn and Zn deprivation in roots and leaves, as shown in Figure 1C,D. In roots, *AhYSL1*, *AhYSL4,* and *AhYSL6* transcript abundances were changed slightly by treatment with various metals, as shown in Figure 1E,G,I. However, in leaves, they were significantly enhanced by excess Cu, as shown in Figure 1F,H,J. Moreover, *AhYSL4* expression was markedly induced by Zn deficiency, and that of *AhYSL6* was suppressed by Cu, Mn, and Zn deficiency in leaves, as shown in Figure 1H,J.

### 3.2. Tissue Expression and Localization of *AhYSL3.1*

The transcript levels of *AhYSL3.1* and *AhYSL3.2* in roots, stems, young and old leaves were determined. Both genes showed similar expression patterns in these tissues, as shown in Figure 2. In roots, *AhYSL3.1* as shown in Figure 2A and *AhYSL3.2* as shown in Figure 2B were expressed in the lateral or main roots, but not the root tips. Among roots, stems, young and old leaves, the expression levels of both genes were highest in young leaves. The tissue localization of *AhYSL3.1* was also checked by in situ hybridization. Regarding tissue localization in the root, *AhYSL3.1* antisense probe staining was more visible around the vascular bundles, as shown in Figure 3C, compared to the sense probe, which was used as a negative control, as shown in Figure 3D.

### 3.3. The Subcellular Localization of *AhYSL3.1*

By the TOPCONS web server, the transmembrane helix positions of AhYSL3.1 were predicted based on different methods. According to the consensus-based methods of TOPCONS, 16 transmembrane domains were found in AhYSL3.1, as shown in Appendix A. The subcellular localization of *AhYSL3.1* was determined by transient expression assay in onion. For subcellular localization, the *AhYSL3.1* open reading frame fused with green fluorescent protein (GFP) was transiently introduced into onion epidermal cells together with 35S-DsRed as a marker of the cytosol and nucleus. AhYSL3.1-GFP fluorescence was observed in the outer region of the cell, but not the cytosol or nucleus, as shown in Figure 3A. In contrast, GFP fluorescence was colocalized with the red signal of 35S-DsRed, as shown in Figure 3B.

### 3.4. *AhYSL3.1* is a Cu–NA Transporter

Because *AhYSL3.1* was markedly induced by Cu deficiency, but not by other metals in roots, we speculated that *AhYSL3.1* transports Cu in addition to the previously identified substrates Fe(III)–DMA and Fe(II)–NA [38]. Therefore, a yeast complementation assay was performed using a Copper Transporter 1 (CTR1) yeast mutant (M10) [48], which is defective in Cu uptake. The ORF of *AhYSL3.1* was subcloned into the yeast expression vector pDR195. When supplied with Cu–NA, expression of *AhYSL3.1* in the mutant significantly improved the growth of the yeast compared with control cells transformed with empty vector, as shown in Figure 4A, suggesting that peanut *AhYSL3.1* is a Cu–NA transporter. In contrast, when supplied with CuSO_4_, the *AhYSL3.1*-expressing yeast mutant did not grow well, as shown in Figure 4B, indicating that *AhYSL3.1* does not transport ionic Cu^2+^.

### 3.5. High-Level Expression of *AhYSL3.1* in Tobacco Results in Tolerance to Excess Cu

*AhYSL3.1* was introduced into tobacco plants and its function investigated. Eighteen transgenic lines were generated. Two lines (T10 and T11) with higher *AhYSL3.1* expression were used for further analysis, as shown in Appendix A. Non-transformed (NT) tobacco plants and the transgenic lines T10 and T11 were treated with various concentrations of Cu. After treatment without Cu or with 25 μM Cu, the fresh weights of NT and transgenic plants showed no difference, as shown in Figure 5A,B,D. When treated with 50 μM Cu, the fresh weight of one line (T10) was significantly higher than that of NT plants, as shown in Figure 5C,D. Moreover, the Cu concentration in young leaves of transgenic lines was markedly lower than in NT plants when treated with 25 or 50 μM excess Cu, as shown in Figure 5E. In old leaves, when treated with 50 μM Cu, the Cu concentration was significantly lower in transgenic plants, as shown in Figure 5F. However, in stems, the Cu concentration was markedly increased in the T10 and T11 lines compared to NT plants grown in the presence of 25 or 50 μM Cu, as shown in Figure 5G, while in roots there was no difference, as shown in Figure 5H. When treated without Cu (0 μM) or with a normal concentration of Cu (0.2 μM), the Cu concentration was similar in NT and transgenic plants in those tissues, as shown in Figure 6E–H. Moreover, the concentrations of Fe, as shown in Appendix A, Mn, as shown in Appendix A, and Zn, as shown in Appendix A, did not differ between the NT and transgenic lines treated with various concentrations of Cu.

### 3.6. Transgenic Rice Plants Are also Tolerant to Excess Cu

Transgenic rice plants with high *AhYSL3.1* expression were generated, as shown in Appendix A. When treated with 100 μM Cu, the transgenic lines grew better than did the NT plants, as shown in Figure 6A. Additionally, the fresh weights of roots, as shown in Figure 6B, and shoots, as shown in Figure 6C, of the transgenic plants were significantly higher than those of NT plants. Moreover, the Cu concentration in new leaves of transgenic rice was markedly decreased compared to NT plants, as shown in Figure 6D. In contrast, in roots, the Cu concentration was enhanced in one of the transgenic plant lines, as shown in Figure 6E. In old leaves, the Cu concentration showed no difference between the NT and transgenic lines, as shown in Figure 6F. 

## 4. Discussion

Cu transporters are key components in the maintenance of Cu homeostasis. Previous results have demonstrated that the CTR and HMA family proteins are important for Cu acquisition and translocation in plants [6,49]. In this study, we characterized one member of the *YSL* gene family involved in Cu transport in peanut.

The *YSL* gene family has been well characterized as an Fe complex transporter in plants. In a previous study, we identified five peanut *YSL* genes. By yeast complementation assay, AhYSL1 was found to specifically transport Fe(III)–DMA, while AhYSL3.1 transports not only Fe(III)–DMA, but also Fe(II)–NA [38]. However, high expression of *AhYSL3.1* in tobacco plants did not affect the Fe concentration in various tissues, but markedly decreased the Cu concentration in leaves subjected to Fe deficiency, as shown in Appendix A. Therefore, we speculate that *AhYSL3.1* may transport Cu in peanut. Further analyses indicated that among five peanut *YSL* genes, only *AhYSL3.1* and *AhYSL3.2* were markedly induced by Cu deficiency, but not by Mn and Zn deficiency, in roots, as shown in Figure 1, suggesting that *AhYSL3.1* and *AhYSL3.2* were likely responsible for Cu translocation.

In *S. cerevisiae*, the CTR-type proteins were identified for copper transport. CTR1 and CTR3 are localized to the plasma membrane and are required for Cu uptake [48,50]. Whereas, CTR2 mediates copper mobilization from vacuoles [51]. In this study, a yeast mutant, which is defective in high-affinity Cu uptake of CTR1, was used for a yeast functional complementation assay. The yeast mutant expressing *AhYSL3.1* grew well in the culture medium containing Cu–NA but not well in the Cu^2+^ condition, as shown in Figure 4, suggesting that in the yeast mutant AhYSL3.1 uptake Cu–NA. This result indicates that AhYSL3.1 is a Cu–NA transporter. Different from *AhYSL3.1*, in *Arabidopsis* COPTs restored the growth of a mutant yeast strain in the Cu^2+^ condition, indicating COPTs transport Cu^2+^ [8].

It is important to note that rice *OsYSL16* has been reported to transport both Cu–NA and Fe(III)-DMA in yeast assay [33,36]. According to a phylogenetic analysis, *AhYSL3.1* belongs to the same group as *OsYSL16* [38]. In contrast to the unchanged expression of *OsYSL16* in Cu-deficient rice roots, *AhYSL3.1* was induced by Cu deficiency in peanut roots, as shown in Figure 1A. Knockout of *OsYSL16* resulted in an increased Cu concentration in older leaves, but decreased Cu concentration in younger leaves in the vegetative stage, suggesting that it functions in delivering Cu to young tissues and seeds through phloem transport [33]. In this study, an artificial promoter driving preferential expression in vascular tissues [43,44] was used to induce exogenous expression of *AhYSL3.1* in tobacco and rice plants. The transgenic tobacco as shown in Figure 5 and rice plants as shown in Figure 6 contained a lower concentration of Cu in young leaves under excess Cu conditions, as shown in Figure 6D, suggesting that *AhYSL3.1* functions in the internal transport of Cu in transgenic plants. The low level of Cu in young leaves of transgenic plants may be attributed to the recycle and export of Cu from young leaves due to the high level of *AhYSL3.1* expression. This speculation can be observed from the higher concentration of Cu in the stem in transgenic tobacco plants, as shown in Figure 5G. There is a model to illustrate the transport of Cu by AhYSL3.1 from leaves to stem in transgenic tobacco, which results in higher concentration of Cu in the stem but lower in the leaves in the excess Cu condition, as shown in Appendix A. In peanut, *AhYSL3.1* was expressed in the main or lateral roots, but not the root tips, as shown in Figure 2A. In situ hybridization suggested that *AhYSL3.1* was localized around the vascular tissues of peanut roots, as shown in Figure 3C,D. Predicted topologies of AhYSL3.1 speculated that it may be a membrane protein, as shown in Appendix A. Moreover, transient expression of *AhYSL3.1* in onion epidermis cells indicated that *AhYSL3.1* was localized to the plasma membrane, as shown in Figure 3A. These results indicated that in peanut plants the Cu–NA transporter AhYSL3.1 mediates internal translocation of Cu.

Consistent with the results for *OsYSL16*, a yeast expression assay indicated that AhYSL3.1 transported Cu–NA, as shown in Figure 4, in addition to Fe(III)–DMA and Fe(II)–NA [38]. Moreover, knockdown of *OsYSL16* resulted in greater Fe accumulation in the vascular bundles of leaves [36], while induction of *OsYSL16* expression resulted in an increased Fe concentration in shoots of *OsYSL16-*induced lines [35], indicating that *OsYSL16* is also responsible for Fe homeostasis. In our study, the Fe concentration was not changed in young leaves or roots of *AhYSL3.1*-overexpressing tobacco plants, as shown in Appendix A, but the Cu concentration in leaves and stems was significantly altered, as shown in Figure 5E–G. Therefore, it is reasonable to speculate that *AhYSL3.1* functions in Cu transport, but not Fe, in peanut plants.

## Figures and Tables

**Figure 1 genes-09-00635-f001:**
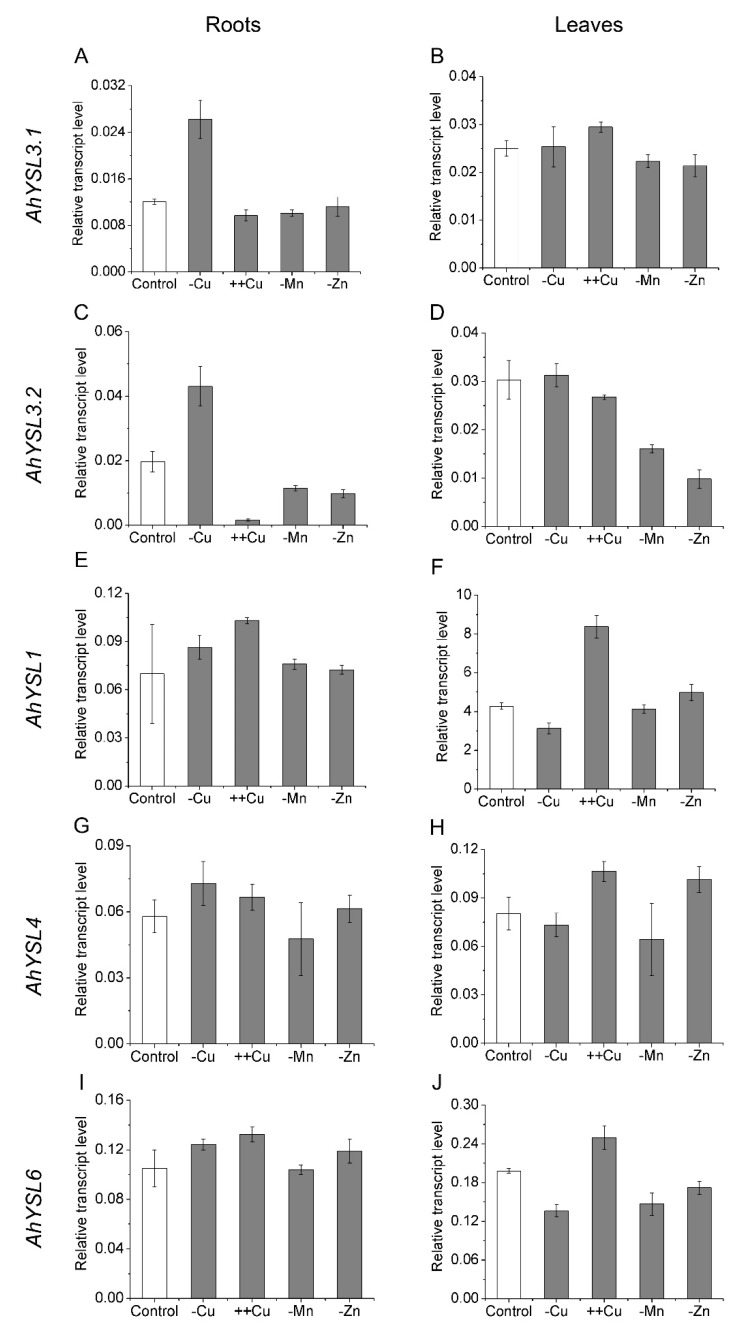
Expression pattern of *AhYSL* genes in response to various metal treatments. Peanut plants were cultured in normal nutrient solution (control) or treated with Cu deficiency (–Cu), excess Cu (++Cu, 25 μM Cu), Mn deficiency (–Mn), or Zn deficiency (–Zn) in hydroponics for 5 days. (**A**,**B**) *AhYSL3.1*. (**C**,**D**) *AhYSL3.2*. (**E**,**F**) *AhYSL1*. (**G**,**H**) *AhYSL4*. (**I**,**J**) *AhYSL6*. Roots (**A**,**C**,**E**,**G**,**I**) and leaves (**B**,**D**,**F**,**H**,**J**) of peanut were sampled using five biological replicates. Vertical bars indicate expression levels relative to the control, *ubiquitin*. Error bars indicate standard deviation.

**Figure 2 genes-09-00635-f002:**
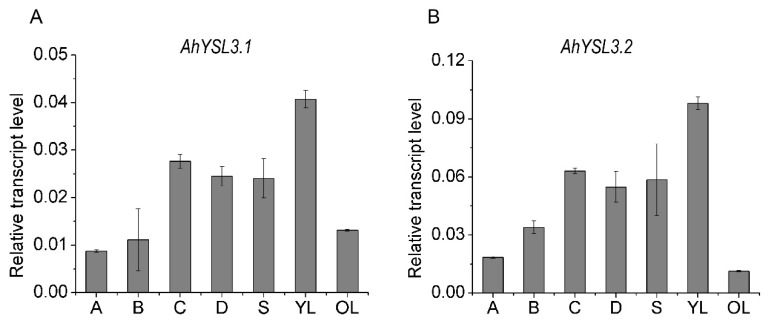
*AhYSL3.1* (**A**) and *AhYSL3.2* (**B**) transcript levels in the indicated parts of peanut plants. The following parts of peanut plants grown under normal conditions for 7 days were harvested: 0–2 cm root tips (A from horizontal abscissa), 2–4 cm root tips (B from horizontal abscissa), lateral roots except the above root tips (C), the main roots (D), stems (S), young leaves (YL), and old leaves (OL). Vertical bars indicate relative expression levels relative to the control, *ubiquitin*. Three biological replicates were performed. Error bars represent standard deviation.

**Figure 3 genes-09-00635-f003:**
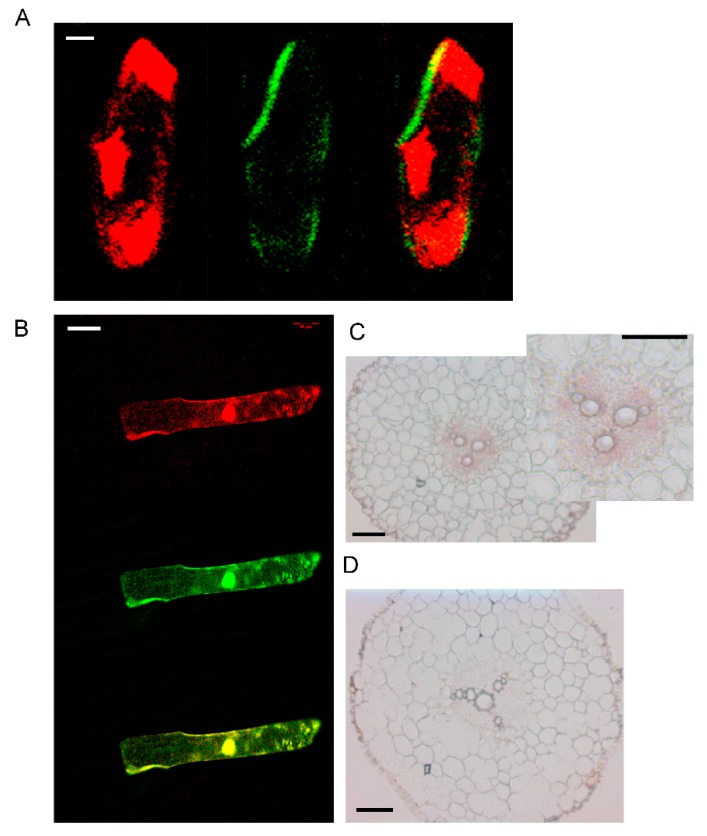
Subcellular and tissue localization of *AhYSL3.1*. (**A**) 35S-DsRed, as a marker of the cytosol and nucleus, was cointroduced into onion epidermal cells with the *AhYSL3.1* ORF fused with GFP. Red signals (left panel) are from 35S-DsRed. Green signals (middle panel) are from AhYSL3.1-GFP. Right panel shows a merged image. Scale bars represent 20 μm. (**B**) GFP alone was colocalized with 35S-DsRed. Scale bars represent 100 μm. (**C**,**D**) In situ hybridization analysis of *AhYSL3.1* in peanut roots. *AhYSL3.1* antisense (**C**) and sense (**D**) probes were hybridized in a cross section of peanut roots. Bars = 100 μm.

**Figure 4 genes-09-00635-f004:**
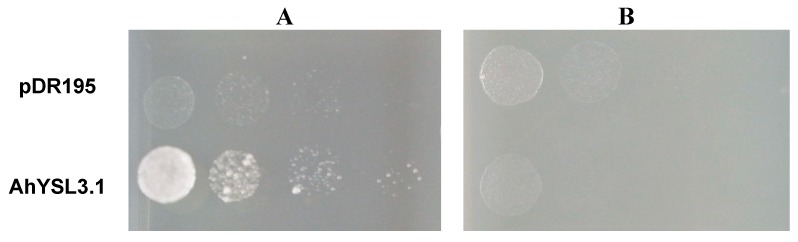
Functional complementation of *AhYSL3.1* in yeast. The yeast strain M10 was transformed with the AhYSL3.1-pDR195 construct or the empty pDR195 vector, which was used as a negative control. Serial dilutions of the transformed yeast cells of OD_600_ 1 to 0.001 were plated onto Synthetic Defined (SD) medium containing 9 μM Cu–NA (**A**) or 10 μM CuSO_4_ (**B**).

**Figure 5 genes-09-00635-f005:**
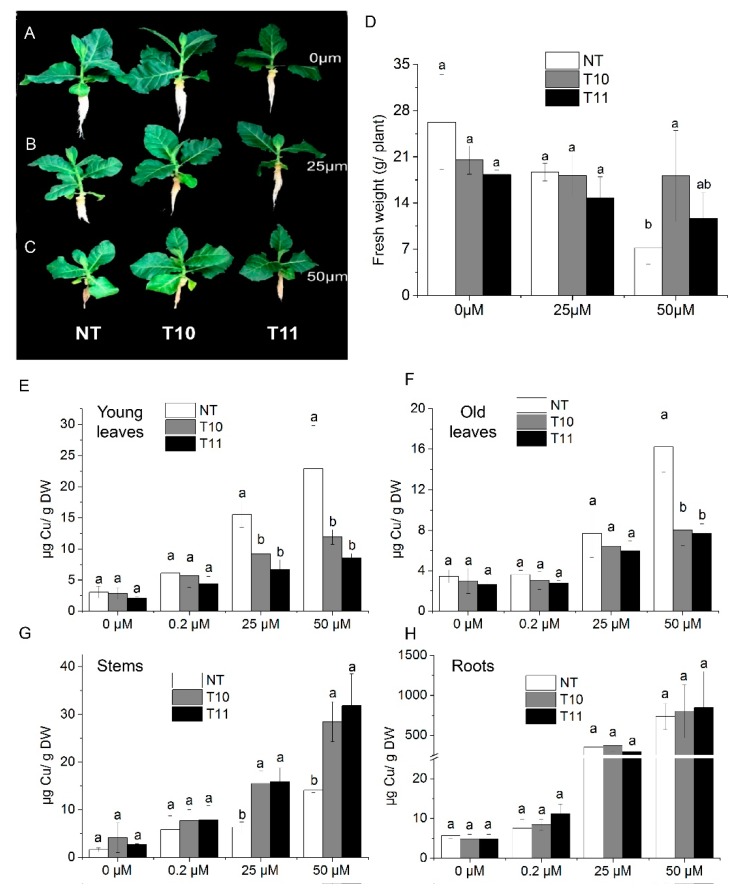
Transgenic tobacco plants expressing *AhYSL3.1* exhibited enhanced tolerance to excess Cu. (**A**–**C**) Phenotypes of transgenic tobacco plants expressing *AhYSL3.1* in response to excess Cu. The transgenic plants were treated with 0 μM (**A**), 25 μM (**B**), or 50 μM (**C**) Cu for 6 days after 1–2 weeks of normal hydroponic culture. NT, non-transformed plants; T10 and T11, transgenic lines. (**D**) Fresh weights of NT and transgenic plants. (**E**–**H**) Cu concentrations in young leaves (**E**), old leaves (**F**), stems (**G**), and roots (**H**) of NT and transgenic lines treated with 0, 0.2, 25, or 50 μM Cu with five biological replicates. Values are means ± SD. Means followed by different letters among NT, T10, and T11 are significantly different according to a least-significant-difference test (*p* < 0.05).

**Figure 6 genes-09-00635-f006:**
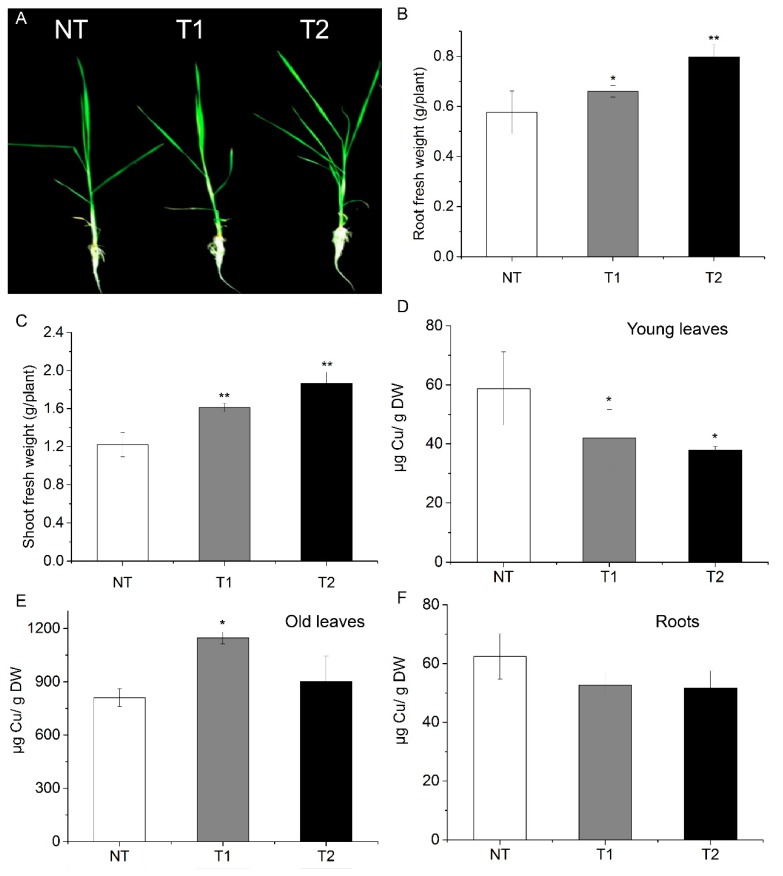
High *AhYSL3.1* expression in rice plants resulted in tolerance to excess Cu. Rice plants were treated with 100 μM Cu for 6 days. (**A**) Phenotypes of transgenic rice plants expressing *AhYSL3.1* in response to excess Cu. NT, non-transformed plants; T1 and T2, transgenic lines. (**B**,**C**) Fresh weights of roots (**B**) and shoots (**C**) of NT and transgenic rice plants after exposure to excess Cu. (**D**–**F**) Cu concentrations in young leaves (**D**), roots (**E**), and old leaves (**F**) of NT and transgenic lines treated with 100 μM Cu. Values are means of four biological replicates. Error bars indicate SD. Significant differences from NT were determined by Student’s *t*-test, * *p* < 0.05, ***p* < 0.01.

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
