# Peer review of "The Yellow Stripe-Like (YSL) Gene Functions in Internal Copper Transport in Peanut"

_genes, 2018, doi:10.3390/genes9120635_

Round 1

Reviewer 1 Report

The manuscript by Dai et al. reports that the yellow stripe-like (YSL) gene functions in copper transport in peanut. Thus, the manuscript contains a novel valuable information. The manuscript is well written and easy to follow. Nonetheless, I have several suggestions, which, in my opinion, would improve the manuscript.

- In my opinion, localization studies should follow in situ hybridization;

- I think additional analysis of AhYSL3.1 domain organization showing the presence of transmembrane segments would support the conclusion that AhYSL3.1 represents plasma membrane protein;  

- Additional statistical analysis can be performed. In addition, I did not find any descriptions of statistical methods used.

- Additional short legends located within the histograms on Fig. 1, 2, 5 and 6 describing experiments would simplify the understanding.

Author Response

The manuscript by Dai et al. reports that the yellow stripe-like (YSL) gene functions in copper transport in peanut. Thus, the manuscript contains a novel valuable information. The manuscript is well written and easy to follow. Nonetheless, I have several suggestions, which, in my opinion, would improve the manuscript.

- In my opinion, localization studies should follow in situ hybridization;

l  According to your suggestion, we have revised the related part. The tissue localization part was combine with tissue expression pattern (section 3.2 page 11 line 4-8). And section 3.3 was renamed “subcellular localization of AhYSL3.1” (page 11 line 10-20)

- I think additional analysis of AhYSL3.1 domain organization showing the presence of transmembrane segments would support the conclusion that AhYSL3.1 represents plasma membrane protein; 

l  According your suggestion, the related result was added in this manuscript (Supplementary Fig S1 page 30).

- Additional statistical analysis can be performed. In addition, I did not find any descriptions of statistical methods used.

l  According to suggestion, the statistical methods were added in section 2.8. (page 10 line3-8)

- Additional short legends located within the histograms on Fig. 1, 2, 5 and 6 describing experiments would simplify the understanding.

l  We have modified the Fig.1, 2, 5 and 6 based on your suggestion.

Reviewer 2 Report

In this manuscript, the authors explain about the Yellow Stripe-like (YSL) gene functions in copper transport in peanut. In its current format, this study cannot be accepted in this journal.

Drawbacks:

1.      No information why use how to generate transgenic lines

2.      What vector use and how many independent lines. Did your line stabilized ? How many copy number? Overexpresion results? Gene functionaling or not? Any downstream genes or target gens? Any mutant?

3.      Have less physiology and morphological data to back theory.

4.      Why use two model system instead of focusing one on rice. These days no one uses tobacco?

5.      No molecular work except some qRT-PCR result.

Author Response

In this manuscript, the authors explain about the Yellow Stripe-like (YSL) gene functions in copper transport in peanut. In its current format, this study cannot be accepted in this journal.

Drawbacks:

1.                     No information why use how to generate transgenic lines

2.                     What vector use and how many independent lines. Did your line stabilized ? How many copy number? Overexpresion results? Gene functionaling or not? Any downstream genes or target gens? Any mutant?

l  The information of related construct for transgenic lines in both rice and tobacco has added in supplementary Fig S2A. For the transgenic tobacco lines, we generated 18 independent lines based on same vector, and we choose two lines (T10 and T11) with higher AhYSL3.1 expression were used for further analysis (Supplementary Fig. S2B). And detail information was described in corresponding caption of supplementary Fig S2A and section 2.6. In this work, we checked the expressed of five YSL family members in response to other metal treatments (Fig. 1/section 3.1). By a yeast complementation assay using a Copper Transporter 1 (CTR1) yeast mutant (M10), we check the function of AhYSL3.1 in Cu transporting (Section 3.4). By analyzing the sequence of AhYSL3.1 in the TOPCONS web server, we found no signal peptide in this gene (supplementary Fig. S1). So it’s hard to find any downstream genes or target genes for a transporter without the related mutant.

3.                     Have less physiology and morphological data to back theory.

l  In this work, we used two transgenic species to test the role of AhYSL3.1 in Cu transporting (Fig. 5 and 6). Based on scientific hypothesis, we checked the Cu concentrations in different tissue of transgenic plants. In our opinion, it is enough to deduce the function of AhYSL3.1 in Cu transporting. 

4.                     Why use two model system instead of focusing one on rice. These days no one uses tobacco?

l  Because YSL family widely exists in strategy Ⅰ and Ⅱ plants. In our previous study, five AhYSL genes were identified in peanut, and their response to Fe deficiency was analyzed (Xiong et al., 2013). Using the two iron-uptake strategy plants will be more convenient to know its function in Fe and other metals, not only Cu. As a strategy Ⅰ plant, tobacco shares the iron uptake strategy with peanut.

5.      No molecular work except some qRT-PCR result.

l  In our opinion, the presented work is enough to deduce the function of AhYSL3.1 in Cu transporting. By conducting more works to confirm its role would take more time and attention.

Round 2

Reviewer 2 Report

Although the results are not sufficient at this stage, it will be hard to get new results, and the author's reply convinces me. It can be accepted in its current format.

Author Response

Although the results are not sufficient at this stage, it will be hard to get new results, and the author's reply convinces me. It can be accepted in its current format.

l  Thanks for your comments and understanding.